# HDAC2- and EZH2-Mediated Histone Modifications Induce PDK1 Expression through miR-148a Downregulation in Breast Cancer Progression and Adriamycin Resistance

**DOI:** 10.3390/cancers14153600

**Published:** 2022-07-23

**Authors:** Yunxia Xie, Zhumei Shi, Yingchen Qian, Chengfei Jiang, Wenjing Liu, Bingjie Liu, Binghua Jiang

**Affiliations:** 1Academy of Medical Science, School of Basic Medical Sciences, Affiliated Cancer Hospital of Zhengzhou University, Zhengzhou University, Zhengzhou 450052, China; xieyunxia@gs.zzu.edu.cn; 2Department of Neurosurgery, The First Affiliated Hospital of Nanjing Medical University, 300 Guangzhou Road, Nanjing 210029, China; shizhumei@njmu.edu.cn; 3Department of Pathology, The Affiliated Jiangning Hospital of Nanjing Medical University, Nanjing 211100, China; qianyingchen@jnyy10.wecom.work; 4Department of Pathology, Nanjing Medical University, 140 Hanzhong Road, Nanjing 210029, China; chengfeijiang@hotmail.com; 5Department of Pathology, Anatomy and Cell Biology, Thomas Jefferson University, Philadelphia, PA 19107, USA; binghua.jiang1@outlook.com

**Keywords:** PDK1, epigenetic modification, miR-148a, breast cancer, Adriamycin resistance

## Abstract

**Simple Summary:**

Epigenetic modification plays an important regulatory role in breast cancer progression. However, the relationship between epigenetic modification with tumor metabolism has not yet been fully elucidated. PDK1, as a key enzyme in glucose metabolism, mediates multiple tumors development. But, the mechanism of epigenetic modification in regulating PDK1 remains unclear in breast cancer. Here, we demonstrated that HDAC2 and EZH2 upregulated PDK1 expression through inhibiting miR-148a. Importantly, miR-148a targeting PDK1 regulated breast cancer cells glycolysis, invasion, epithelial-mesenchymal transition (EMT) and Adriamycin resistance. Our results suggested that the HDAC2/EZH2/miR-148a/PDK1 axis may be a promising potential therapeutic strategy.

**Abstract:**

Background: Breast cancer has one of highest morbidity and mortality rates for women. Abnormalities regarding epigenetics modification and pyruvate dehydrogenase kinase 1 (PDK1)-induced unusual metabolism contribute to breast cancer progression and chemotherapy resistance. However, the role and mechanism of epigenetic change in regulating PDK1 in breast cancer remains to be elucidated. Methods: Gene set enrichment analysis (GSEA) and Pearson’s correlation analysis were performed to analyze the relationship between histone deacetylase 2 (HDAC2), enhancer of zeste homologue 2 (EZH2), and PDK1 in database and human breast cancer tissues. Dual luciferase reporters were used to test the regulation between PDK1 and miR-148a. HDAC2 and EZH2 were found to regulate miR-148a expression through Western blotting assays, qRT-PCR and co-immunoprecipitation assays. The effects of PDK1 and miR-148a in breast cancer were investigated by immunofluorescence (IF) assay, Transwell assay and flow cytometry assay. The roles of miR-148a/PDK1 in tumor growth were investigated in vivo. Results: We found that PDK1 expression was upregulated by epigenetic alterations mediated by HDAC2 and EZH2. At the post-transcriptional level, PDK1 was a new direct target of miR-148a and was upregulated in breast cancer cells due to miR-148a suppression. PDK1 overexpression partly reversed the biological function of miR-148a—including miR-148a’s ability to increase cell sensitivity to Adriamycin (ADR) treatment—inhibiting cell glycolysis, invasion and epithelial–mesenchymal transition (EMT), and inducing apoptosis and repressing tumor growth. Furthermore, we identified a novel mechanism: DNMT1 directly bound to EZH2 and recruited EZH2 and HDAC2 complexes to the promoter region of miR-148a, leading to miR-148a downregulation. In breast cancer tissues, HDAC2 and EZH2 protein expression levels also were inversely correlated with levels of miR-148a expression. Conclusion: Our study found a new regulatory mechanism in which EZH2 and HDAC2 mediate PDK1 upregulation by silencing miR-148a expression to regulate cancer development and Adriamycin resistance. These new findings suggest that the HDAC2/EZH2/miR-148a/PDK1 axis is a novel mechanism for regulating cancer development and is a potentially promising target for therapeutic options in the future.

## 1. Background

Breast cancer is a multifaceted heterogeneous disease and the leading cause of cancer-related death in women [1]. Although cancer treatment has greatly improved survival, chemoresistance and metastasis remain major hurdles for breast cancer therapy. Adriamycin (ADR) is widely used as a momentous chemotherapy agent for treating breast cancer. However, one of the major obstacles to its treatment success is the acquisition of ADR resistance [2].

The Warburg effect describes the well-recognized metabolic phenotype that tumor cells heavily depend on glycolysis for energy metabolism [3]. The Warburg effect is a biochemical hallmark of tumors [4]. The abnormal expression of glycolytic pathway-related proteins, including glucose transporters (GLUTs), hexokinase (HK), phosphofructokinase (PFK), and pyruvate kinase (PK), is associated with the malignant progression of various human cancers [5,6]. In breast cancer, coactivator-associated arginine methyltransferase 1 (CARM1) methylates pyruvate kinase M2 isoform (PKM2) to accelerate aerobic glycolysis, promoting breast tumorigenesis [7]. Glycolysis gatekeeper pyruvate dehydrogenase kinase 1 (PDK1) has been shown to inhibit pyruvate dehydrogenase through phosphorylation and prevent pyruvate from entering the tricarboxylic acid (TCA) cycle, which plays a pivotal role in the process of facilitating glycolysis [8]. PDK1 has been reported to promote tumor cell migration and proliferation, and to regulate vessel growth in the colon tumor microenvironment [9,10]. Moreover, PDK1 may be involved in breast cancer metastases to the liver [11,12]. However, the role and mechanism of PDK1 in breast cancer development remains to be elucidated.

Accumulating evidence suggests that epigenetic alterations contribute to cancer initiation and progression [13,14]. Histone modification (such as acetylation or methylation on the lysine residue) is a major form of epigenetic change in the promoter regions of genes to regulate gene expression [15]. Histone deacetylases (HDACs) are a class of chromatin-modulating enzymes that remove acetyl groups from histones and regulate the expression of numerous proteins involved in cancer progression [16,17]. HDAC inhibitors blunt glycolysis in glioblastoma [18]. HDAC3 is involved in aerobic glycolysis through miR-31 [19]. Enhancer of zeste homologue 2 (EZH2) is the catalytic subunit of the polycomb repressive complex 2 (PRC2), which suppresses gene expression by methylating histone H3 lysine 27 [20]. The PRC2-mediated inhibition of gluconeogenic enzyme fructose-1,6-bisphosphatase is important in liver and kidney carcinogenesis [21]. EZH2 regulated STAT3 and FoxO1 signaling to promote tumor glycolysis in human oral squamous cell carcinoma cells [22]. In breast cancer, a few studies have demonstrated that EZH2 and HDACs play an indispensable role in breast cancer progression [23,24].

To clarify the mechanism of epigenetic modification in regulating PDK1 in breast cancer, we address following questions: Firstly, to explore whether PDK1 expression is dysregulated in breast cancer and the aberrant expression of PDK1 is regulated by histone deacetylase 2 (HDAC2) and EZH2. Secondly, to verify whether PDK1 is a directly functional target gene of miR-148a. Thirdly, to elucidate the molecular mechanism of miR-148a suppression by the epigenetic modification of enzymes HDAC2 and EZH2. Finally, to verify whether miR-148a and PDK1 are involved in cell chemoresistance, metastasis, proliferation, and apoptosis in breast cancer cells. In this study, we elucidate that these epigenetic alterations play a vital role in regulating PDK1 and identify the new regulation axis of EZH2/HDAC2/miR-148a/PDK1 in breast cancer progression and therapeutic resistance.

## 2. Methods

### 2.1. Cell Lines and Reagents

Breast cancer cell lines (MCF10A (RRID: CVCL_0598), MCF7 (RRID: CVCL_0031), MDA-MB-231 (RRID: CVCL_0062)), HEK293T (RRID: CVCL_0063) and A549 (RRID: CVCL_0023) were purchased from American Type Culture Collection (ATCC, Gaithersburg, MD, USA). MCF7/ADR cell line, A2780, and U251 cell lines were obtained from previous studies [25,26]. The cells were cultured in Dulbecco’s Modified Eagle’s Medium (DMEM) basic culture medium (Gibco Life Technology, New York, NY, USA) supplemented with 10% fetal bovine serum (Gibco Life Technology, New York, NY, USA) and 1% penicillin streptomycin (Gibco Life Technology, New York, NY, USA) in an incubator with 5% CO_2_ at 37 °C. HDAC2 siRNA and EZH2 siRNA (ENX-1) were bought from Santa Cruz Biotechnology (Cat# sc-29345, Cat# sc-35312, Dallas, TX, USA). 3-Deazaneplanocin A (DZNep) was purchased from MedChemExpress company (Cat# 120964-45-6, Monmouth Junction, NJ, USA). PDK1 (RRID: Addgene_20564) was purchased from Addgene company (Watertown, MA, USA). Glucose and lactate levels were measured by glucose assay kit and lactate assay kit (BioVision, Milpitas, CA, USA).

### 2.2. Tissue Samples

In this study, breast tumor tissues and adjacent normal breast tissues of breast cancer patients were obtained from the tissue bank of Zhengzhou University (Zhengzhou, China). These tissues had been kept in the tissue bank for several years. These breast cancer patients did not receive surgical treatment, radiotherapy, chemotherapy or immunotherapy before the surgery. The patient information for the breast cancer tissues used in this study was listed in Appendix A.

### 2.3. Lentiviral Packaging and Stable Cell Line Establishment

For transfection, 4 μg lentivirus plasmids carrying miR-NC, miR-148a or miR-148a + PDK1, 3 μg psPAX2 and 1 μg pMD2.G plasmids (RRID: Addgene_12260, RRID: Addgene_12259, Addgene, Watertown, MA, USA) were co-transfected into 293T cells. After 24 h, the cell culture supernatant was collected, then used to infect MCF7 and MDA-MB-231 cells. There was a red fluorescent protein (RFP) tag on the lentiviral vector, carrying miR-NC or miR-148a, which was used to check the infection efficiency using microscopy and cells were selected by puromycin.

### 2.4. RNA Extraction and qRT-PCR Analysis

The total RNAs from cells and tissues were extracted using the RNAiso kit (Takara, Dalian, China). qRT-PCR assay was carried out to measure the mRNA expression levels of genes in this study according to the manufacturer’s instructions (Takara, Dalian, China) using an ABI Prism 7900 HT instrument, U6 as the miRNA control gene, and β-actin as the mRNA control gene. The primers were as follows: β-actin forward primer, 5′-ACCGGGCATAGTGGTTGGA-3′; β-actin reverse primer, 5′-ATGGTACACGGTTCTCAACATC-3′; PDK1 forward primer, 5′-CTGTGATACGGATCAGAAACCG-3′; PDK1 reverse primer, 5′-TCCACCAAACAATAAAGAGTGCT-3′; EZH2 forward primer, 5′-AATCAGAGTACATGCGACTGAGA-3′; EZH2 reverse primer, 5′-GCTGTATCCTTCGCTGTTTCC-3′; HDAC2 forward primer, 5′-ATGGCGTACAGTCAAGGAGG-3′; HDAC2 reverse primer, 5′-TGCGGATTCTATGAGGCTTCA-3′. miRNA quantification: Bulge-loopTM miRNA qRT-PCR primer sets (one RT primer and a pair of qPCR primers for each set) specific for U6 and miR-148a-3p were designed by the RiboBio company (Guangzhou, China).

### 2.5. Western Blotting Analysis

Cells were lysed using cell lysis solution (Beyotime Biotechnology, Shanghai, China) containing protease inhibitors (Thermo Scientific, Waltham, MA, USA). The total proteins were quantified using a protein quantification kit (Thermo Scientific, Waltham, MA, USA). The proteins were separated on 10% gradient sodium dodecyl sulfate-polyacrylamide gel electrophoresis (SDS-PAGE) gels. The primary antibodies were used as follows: EZH2 (Cat# 4905, Polyclonal, RRID: AB_2278249, 1:1000), HDAC2 (D6S5P) (Cat# 57156, Monoclonal, 1:1000), Cytochrome C (Cat# 4272, Polyclonal, RRID: AB_2090454, 1:1000), Bcl-2 (Cat# 2872, Polyclonal, RRID: AB_10693462, 1:1000), Caspase-3 (Cat# 9662, Poly-clonal, 1:1000), and cleaved Caspase-3 (Asp175) (Cat# 9661, Polyclonal, RRID: AB_2341188, 1:1000) primary antibodies were purchased from Cell Signaling Technology (Danvers, MA, USA). PDK1 (Cat# 18262-1-AP, Polyclonal, 1:1000), β-actin (Cat# 66009-1-Ig, Monoclonal, CloneNo. 2D4H5, 1:5000), BAX (Cat# 60267-1-Ig, Monoclonal, CloneNo. 4G5E8, 1:2000), and Vimentin (Cat# 60330-1-Ig, Monoclonal, CloneNo. 3H9D1, 1:2000) primary antibodies were purchased from Proteintech company (Wuhan, China). E-cadherin (Cat# BS1097, Polyclonal, 1:500) was bought from the Bioworlde company (St. Louis Park, MN, USA). Protein bands were incubated with horseradish peroxidase-conjugated secondary antibodies and visualized with a chemiluminesence detection reagent (Advansta, San Jose, CA, USA).

### 2.6. Immunohistochemistry Assay

Tumor tissues of different groups were fixed with 4% paraformaldehyde for 24 h, subsequently dehydrated by an ethanol gradient, and embedded in paraffin. Paraffin-embedded tissues were then cut into 5 μm sections and baked at 65 °C for 40 min. These tumor sections were dewaxed, then antigen repair was carried out with 0.01 M sodium citrate solution. The tumor sections were blocked with 1% BSA for 1 h at room temperature, and incubated with primary antibodies (PDK1 (1:200), Bcl-2 (1:600) and Vimentin (1:2000)) at 4 °C overnight. Finally, the sections were incubated with secondary antibodies and stained with the DAB Histochemistry Kit (Invitrogen, Waltham, MA, USA).

### 2.7. Co-Immunoprecipitation Assay

MDA-MB-231 cells were individually transfected with EZH2 (Cat# 173717, RRID: Addgene_173717, Watertown, MA, USA), HDAC2 overexpression plasmids (Customized by GenePharma, Shanghai, China), and their respective control plasmids. After 48 h, total proteins were extracted by cell lysis buffer. First, preprocessed magnetic beads (Bimake, Houston, TX, USA) were incubated with antibody solution for 15 min according to the manufacturer’s instructions. After washing the crosslinked products, protein supernatants were added to crosslinked products and incubated at 4 °C overnight. Finally, binding proteins were eluted and detected by Western blotting as previously described.

### 2.8. Luciferase Assay

PDK1-3′UTR-WT- and PDK1-3′UTR-MUT-sequences were constructed in pmiRGLO plasmids, respectively (obtained from previous study [25]). The plasmids were co-transfected with miR-148a mimics or miR-NC into HEK293T cells. After 48 h, the cells were lysed. The *Firefly luciferase* (*luc2*) and *Renilla luciferase* (*hRluc-neo*) activities were detected by the Dual-Luciferase^®^ Reporter Assay System kit (Promega, Madison, WI, USA). Plasmid construction primers: PDK1-3′UTR-WT-pmiRGLO forward primer, 5′-GAGTA CTCATGTCTCACCTAACCCAC-3′; PDK1-3′UTR-WT-pmiRGLO reverse primer, 5′-CCCTCGAGGGATCACAGTGCAAGAAAGT-3′; PDK1-3′UTR-MUT-pmiRGLO forward primer, 5′-GAGTACTCATGTCTCACCTAACCCAC-3′; PDK1-3′UTR-MUT-pmiRGLO reverse primer, 5′-CCCTCGAGGGATCACACACCAAGAAAGT-3′.

### 2.9. Transwell Assay

Transwell chambers with 8 μm pores were purchased from Corning company (New York, NY, USA). The cells of MDA-MB-231 miR-NC, MDA-MB-231 miR-148a, and MDA-MB-231 miR-148a + PDK1 were digested with 0.25% trypsin. Then, 3 × 10^5^ cells were resuspended in 200 μL serum-free medium and placed in the upper chamber. In the lower chamber, 500 μL DMEM with 10% fetal bovine serum was added. After incubating at 37 °C for 24 h, the cells were fixed with 10% methanol and stained with 0.05% crystal violet for 20 min. The cells on the upper side of the membrane were gently dislodged. The cells on the lower side were observed using a microscope (200×). Next, 200 μL 33% acetic acid was added to the lower chamber, which was gently shaken for 20 min. The crystal violet was eluted, and the absorbance values at OD 570 nm were measured.

### 2.10. Immunofluorescence Assay

MDA-MB-231 miR-NC, MDA-MB-231 miR-148a, and MDA-MB-231 miR-148a + PDK1 (1 × 10^5^ cells) were seeded in Millicell EZ 8-well slides (Millipore, Burlington, MA, USA). After 24 h, the cells were fixed with 4% formaldehyde and permeabilized with PBST containing 0.5% Triton X-100 for 20 min and blocked with 1% BSA/PBST for 2 h at room temperature. Next, 1% BSA/PBST containing antibodies against Vimentin (1:100) and E-cadherin (1:100) were added to the wells, and the slides were incubated at 4 °C overnight. Fluorescein isothiocyanate (FITC)-labeled goat anti-rabbit secondary antibodies (1:200, Santa Cruz Biotechnology, Dallas, TX, USA) and tetramethyl rhodamine isothiocyanate (TRITC)-labeled goat anti-mouse secondary antibodies (1:200, Santa Cruz Biotechnology, Dallas, TX, USA) were used together to detect the two target proteins. Different colors of fluorescence were observed under a fluorescence microscope (ZEISS, Oberkochen, Baden-Württemberg, Germany).

### 2.11. MMP Detection Assays

The cells of MDA-MB-231 miR-NC, miR-148a, miR-NC + PDK1 and miR-148a + PDK1 were seeded into a 6-well plate. When the cell density reached 70%, 1 mL JC-1 solution (Beyotime Biotechnology, Shanghai, China) was added to the wells, and the plate was incubated at 37 °C for 20 min. After washing twice with JC-1 buffer, the levels of JC-1 aggregates and monomer were observed under a fluorescence microscope. The levels of JC-1 aggregates and monomer in each group were quantified by using Image J software (v1.8.0).

### 2.12. Annexin-V/PI Double-Staining Assay

We used the Annexin V-FITC apoptosis assay kit (univ company, abs50001, Shanghai, China) to detect cell apoptosis according to the manufacturer’s protocol. First, we collected 1 × 10^5^ cells with different treatment. Then, 10× binding buffer was diluted to 1× binding buffer and then was used to suspend the cells. Second, the cells were incubated with 3 µL Annexin V-FITC and 5 µL propidium iodide (PI) for 15 min in the dark. Finally, the apoptosis levels of the samples were detected by flow cytometry (BD Biosciences, San Jose, CA, USA). Annexin V-FITC: maximum excitation light was 488 nm and emission light was 520 nm. PI: maximum excitation light was 535 nm and emission light was 617 nm.

### 2.13. Tumorigenesis Study

Five-week-old BALB/c female nude mice were purchased from the Shanghai Laboratory Animal Center (Chinese Academy of Sciences, Shanghai, China) and were randomly divided into three groups with five mice in each group. MDA-MB-231 cells (5 × 10^6^ cells) expressing miR-NC, miR-148a, and miR-148a + PDK1 were trypsinized and resuspended in 100 μL serum-free medium and subcutaneously injected into the backs of the mice. Ten days later, we started to measure the tumor volumes every two days. After three weeks, the mice were euthanized and the xenograft tumors were harvested. Animal handling and experimental procedures were performed with the approval of the Institutional Committee on the Animal Care of Nanjing Medical University (Ethics No. IACUC1601177).

### 2.14. GSEA

The BRCA mRNA dataset, miRNA dataset and clinical information were downloaded from TCGA. The mRNA dataset was split into two groups (miR-148a high-expression group/miR-148a low-expression group or PDK1 high-expression group/PDK1 low-expression group) according to the miR-148a or PDK1 median value of all sample expression levels. The GSEA program was used to analyze the enrichment pathways of these groups [27].

### 2.15. Correlation Analysis

The GSE27447 dataset was subjected to normalization (RMA method) and then samples underwent Pearson’s correlation analysis with the *corrplot* package in R software (R Studio, Washington, DC, USA) (R version 3.6.1). The breast cancer data from TCGA were used to perform the Pearson’s correlation analysis between PDK1 and HDACs/PRC2 members using *the GGally* and *ggplot2* package in R software (R version 3.6.1) [28].

### 2.16. Statistical Analysis

The data are represented as the mean ± SEM or SD from at least three independent experiments. Data were analyzed using GraphPad Prism 5 (La Jolla, CA, USA). Pearson’s analysis was used for correlation analysis. Quantitative variables were analyzed using Student’s unpaired two-tailed *t*-test. For human tissue samples, gene expression levels were analyzed using Student’s paired two-tailed *t*-test. Tumor volumes were compared by the analysis of variance of repeated measurements. *p* values at *p* < 0.05 were considered to have statistically significant difference.

## 3. Results

### 3.1. PDK1 Levels Were Upregulated by HDAC2 and EZH2 Which Are Associated with Glycolysis

To study the role of PDK1 in breast cancer, we showed that PDK1 levels were significantly higher in breast cancer tissues than normal tissues using the GSE22820 breast cancer dataset for normal (*n* = 10) and breast cancer tissues (*n* = 176) (Figure 1A). Similar results were obtained in the GSE37751 dataset (Appendix A). Furthermore, we analyzed 3550 breast cancer samples in the Kaplan–Meier plotter (http://kmplot.com/analysis/index.php?p=service&cancer=breast (accessed on 14 August 2021)) and showed that low PDK1 expression group had a longer relapse-free survival time than the group with higher PDK1 levels (Figure 1B). We found that levels of PDK1 in MCF7 and MDA-MB-231 breast cancer cells were significantly higher than those in immortalized normal MCF10A breast cells (Figure 1C).

Epigenetic modification plays an important regulatory role in tumorigenesis and development. To further explore whether histone modification was involved in the regulation of PDK1 expression, we carried out a correlation analysis between levels of PDK1 and HDACs in breast cancer samples from the TCGA dataset. Our results showed that HDAC2 levels were mostly positively correlated with PDK1 levels and high HDAC2 expression had a poor overall survival probability (Figure 1D and Appendix A). In addition, we also performed a correlation analysis between levels of PDK1 and the core components of PRC2 including EZH2, SUZ12, and EED. The analysis results suggested that EZH2, SUZ12 and EED were all significantly associated with PDK1 levels (Figure 1E). However, the Kaplan–Meier plotter (http://kmplot.com/analysis/ (accessed on 14 August 2021) indicated that patients with high EZH2 expression, but not EED, had a poor overall survival (OS) probability (Appendix A). We also found that the expression levels of SUZ12 in Her2-positive samples and triple-negative samples did not have significant difference compared to the normal samples (online website: UALCAN, Appendix A). However, the levels of EZH2 and HDAC2 increased with the increase in the malignancy of breast cancer (Appendix A). The expression levels of HDAC2 and EZH2 were significantly higher in multiple tumor types, indicating that HDAC2 and EZH2 may play important roles as oncogenes (Appendix A). We further measured the protein levels of HDAC2 and EZH2 in MCF10A, MCF7, and MDA-MB-231 cell lines by Western blotting, and showed that MCF7 and MDA-MB-231 cells had higher expression levels of HDAC2 and EZH2 than MCF10A cells (Figure 1F).

To evaluate the correlation between PDK1 levels and glycolysis, we analyzed the enrichment scores of glycolysis-related pathways using the TCGA dataset and the GSEA program. The results showed that PDK1 expression levels were enriched in the HALLMARK_GLYCOLYSIS pathway (NES = 1.65, FDR q = 0.041) (Figure 1G). Similarly, the expression levels of HDAC2 and EZH2 were positively correlated with cellular glycolysis levels (NES = 1.67, FDR q = 0.04; NES = 1.78, FDR q = 0.024) (Figure 1H,I). We further compared the expression levels of PDK1 in HDAC2 and EZH2 low- and high-expression groups with the median value as the grouping standard of HDAC2 and EZH2. The results showed that the expression levels of PDK1 were positively correlated with HDAC2 and EZH2 expression levels (Appendix A). To test the direct regulatory link, we knocked down HDAC2 and EZH2 expression using siRNAs in the cancer cell lines, and found that EZH2 and HDAC2 silencing in the cells significantly decreased PDK1 expression (Figure 1J). Taken together, these results imply that HDAC2 and EZH2 expression levels were highly correlated with PDK1 levels in the human cohort and HDAC2 and EZH2 upregulated PDK1 expression.

### 3.2. miR-148a Was Identified for Directly Suppressing PDK1 Expression and Inhibiting Glycometabolism

To further understand the mechanism of PDK1 upregulation, we found that at the post-transcript level, PDK1 was a potential target of miR-148a and constructed luciferase reporters with the miR-148a wild-type binding sites of the PDK1 3′-UTR regions and with the mutation miR-148a binding sites of the three nucleotide substitutes (Figure 2A). Dual-luciferase reporter assay with miR-148a mimic or miR-NC showed that the luciferase activities of the PDK1 wild-type (WT) reporter were significantly decreased by miR-148a mimic compared to that by miR-NC, while no significant differences in the luciferase activities of the PDK1 mutation (MUT) reporter were observed by the miR-148a mimic (Figure 2B). Both PDK1 mRNA and protein expression levels were attenuated in miR-148a overexpression cells analyzed using qRT-PCR and Western blotting (Figure 2C,D and Appendix A). To further investigate whether miR-148a regulated glycometabolism through PDK1, the culture media from the negative control cells and cells stably expressing miR-148a and miR-148a/PDK1 co-overexpression were collected to analyze the levels of glucose and lactate. The results showed that PDK1 forced expression reversed the miR-148a-inhibiting effect of glucose consumption and lactate production (Figure 2E,F) in the cells. These results indicated that miR-148a directly targeted and suppressed PDK1 expression in the glycolysis regulation process.

### 3.3. HDAC2 Overexpression Suppressed miR-148a Expression

Epigenetic modification and miR-148a have important regulatory roles on PDK1; in this way, how these two factors control the specific mechanism of the abnormal expression of PDK1 in breast cancer needs to be further explored. Our previous studies found that miR-148a expression is inhibited by DNMT1 [29]. It has been reported that not only DNA methylation modification, but histone modification, including acetylation and methylation, may play roles in miRNA expression regulation [30]. Thus, we hypothesized that epigenetic modification regulates the expression of miR-148a, and finally that miR-148a directly combines with PDK1 to regulate the development of breast cancer.

To verify the hypothesis, we measured the expression levels of miR-148a in MCF7 and MDA-MB-231 cells treated with Trichostatin A (TSA), and found that HDAC inhibitor TSA significantly increased miR-148a expression (Figure 3A). We further analyzed the independent breast cancer gene expression dataset (GSE27447) from the public GEO database and were surprised to find that the strongest negative correlation was between HDAC2 and miR-148a (Figure 3B) (*r* = −0.52, *p* < 0.05). HDAC2 protein levels significantly decreased in TSA-treated cells with Western blotting (Appendix A). Moreover, the qRT-PCR results showed that the expression levels of miR-148a were significantly decreased when HDAC2 was overexpressed, whereas the knockdown of HDAC2 using siRNA increased miR-148a expression in MCF7 and MDA-MB-231 cells (Figure 3C,D).

### 3.4. EZH2 Overexpression Inhibited miR-148a Expression through the Recruitment of HDAC2

To further explore whether histone methylation also played a regulatory role in miR-148a expression, we used PRC2 inhibitor 3-deazaneplanocin A (DZNep) to treat MCF7 and MDA-MB-231 cells, and showed that the miR-148a expression levels were significantly increased in the treated cells compared to the control (Figure 4A). DZNep treatment decreased the expression levels of EZH2, which is a core histone methylase of PRC2 (Appendix A). Using the GSE27447 dataset, we further analyzed the correlation between miR-148a and histone methylation modification proteins, and found that there was a negative correlation between EZH2 and miR-148a expression levels (Figure 4B) (*r* = −0.37, *p* = 0.1285). To test whether there was a direct link of EZH2 to the regulation of miR-148a expression, EZH2 was overexpressed in the cells. The upregulation of EZH2 significantly repressed miR-148a expression (Figure 4C). Similarly, the inhibition of EZH2 by siRNAs increased miR-148a expression (Figure 4D). When HADC2 and EZH2 were inhibited by siRNAs in human lung cancer cell line A549 (Appendix A), miR-148a expression levels also significantly increased (Appendix A). Similar results were obtained using human glioma cell line U251 (Appendix A). These findings suggest that HDAC2 and EZH2 repressed miR-148a expression in multiple cancer cell types. To further investigate the regulatory mechanism of HDAC2 and EZH2, the cells were transfected with control, HDAC2, or EZH2 plasmid to study the protein–protein interactions. The co-immunoprecipitation (Co-IP) assay results illustrated that HDAC2 bound to EZH2 and formed an inhibitor complex (Figure 4E). Our previous study evidenced that DNMT1 binds to the promoter of miR-148a and regulates its expression [29]. We further explored these potential relationships using the STRING database to find links between DNMT1, EZH2 and HDAC2 proteins. The analysis results showed that they were strongly interrelated (Appendix A). In addition, we analyzed the methylation modification sites and the histone acetylation modification sites of the miR-148a promoter sequence using the UCSC database, and found that there was a significant overlap among them (Appendix A). Based on these results, a Co-IP assay was performed, and we demonstrated that DNMT1 directly bound with EZH2 rather than HDAC2 (Figure 4F). In summary, DNMT1 directly combined with EZH2 then recruited EZH2 and HDAC2 to the promoter region of miR-148a to form a complex to inhibit miR-148a expression.

### 3.5. The Levels of HDAC2 and EZH2 Were Inversely Correlated with miR-148a Levels in Breast Cancer Tissues

To further test the regulation of miR-148a with HDAC2 and EZH2, we analyzed the expression levels of HDAC2, EZH2, and miR-148a in twenty pairs of human breast cancer tissues and their corresponding normal breast tissues. The qRT-PCR results showed that miR-148a levels were downregulated in breast cancer tissues (Figure 5A). Significantly higher expression levels of HDAC2 and EZH2 were detected in breast cancer tissues than those in normal tissues (Figure 5B), which was consistent with the IHC results for breast cancer and normal breast tissues from the Human Protein Atlas Database [31] (Figure 5C). Pearson’s correlation analysis indicated that HDAC2 and EZH2 protein expression levels were reversely correlated with miR-148a expression (Figure 5D,E).

### 3.6. PDK1 Overexpression Reversed the Effect of miR-148a Decreasing Adriamycin Resistance

Recent research has indicated that PDK1 contributes to chemotherapy resistance [32,33]. However, the role of miR-148a/PDK1 for therapeutic effects in breast cancer is unclear. We found that Adriamycin-resistant breast cancer cell line MCF7/ADR not only had lower levels of miR-148a expression, but also higher levels of PDK1 expression compared to the control (MCF7) (Appendix A). We speculated that miR-148a increased breast cancer cell treatment response to Adriamycin through PDK1. First, we identified that miR-148a overexpression significantly decreased the IC50 value (0.604 vs. 0.305) in the MDA-MB-231/miR-148a group under Adriamycin treatment by using Cell Counting Kit (CCK-8) assays (Figure 6A). Similarly, there was also a significantly lower IC50 value (0.637 vs. 0.324) in the MCF7/miR-148a group (Appendix A). We further found that under the Adriamycin treatment, the proliferative abilities of the MDA-MB-231/miR-148a and MCF7/miR-148a cell lines were significantly lower, while PDK1 overexpression partially restored cell viability (Figure 6B and Appendix A). We showed that miR-148a overexpression increased the apoptosis levels in Adriamycin-treated MDA-MB-231 and MCF7 cells, while PDK1 overexpression abrogated miR-148a-induced apoptosis (Figure 6C and Appendix A). Meanwhile, the apoptosis-related proteins levels of BAX and cleaved caspase-3 increased, whereas Bcl-2 expression levels decreased in miR-148a overexpression cells, and PDK1 overexpression partly reversed these effects caused by miR-148a (Figure 6D and Appendix A). In addition, Western blotting results showed that HDAC2 and EZH2 levels were significantly higher in the Adriamycin-resistant cell line compared with the parental MCF7 cells (Appendix A). These findings suggest that PDK1 overexpression reversed the result of miR-148a increasing the sensitivity of breast cancer cells to Adriamycin treatment, which resulted in a decrease in apoptosis and the promotion of cell proliferation.

It is known that the destruction of mitochondrial membrane potential (MMP) is one of the necessary conditions for cell apoptosis [34]. To further explore whether miR-148a/PDK1 mediates the process of apoptosis by MMP, we used the mitochondrial membrane potential probe JC-1 to detect MMP in MDA-MB-231 stable cell lines. By comparing the ratio of JC-1 aggregates (red) and monomer (green), we found that PDK1 inhibited mitochondrion damage by maintaining MMP and that miR-148a significantly decreased the MMP, triggered apoptosis, which was restored by PDK1 (Figure 6E). We further used ImageJ (image processing tools) to more intuitively quantify the results of the JC-1 aggregates and monomers, as shown in Figure 6E, by calculating the mean optical density (MOD) of the image (Figure 6F). The decrease in MMP caused mitochondrial damage, and thereby induced Cytochrome C flow to the cytoplasm from the mitochondria. We also detected the levels of Cytochrome C in the cytoplasm by Western blotting. The results showed that the levels of Cytochrome C were much higher in the miR-148a group than the miR-NC group, which was partly reversed by PDK1 overexpression (Figure 6G). These results showed that the inhibition of PDK1 by miR-148a increased the response of breast cancer cells to Adriamycin treatment by inducing cell apoptosis.

### 3.7. PDK1 Suppression by miR-148a Inhibited Epithelial–Mesenchymal Transition and Cell Migration

To further study the roles of PDK1 and miR-148a in breast cancer development, we carried out GSEA analysis in breast cancer samples from the TCGA dataset, and found that PDK1 was positively correlated with TGF-BETA signaling (NES = 1.55, *p* value < 0.05, FDR q < 0.2) (Figure 7A). The TGF-β signaling pathway has been reported to be involved in the regulation of epithelial–mesenchymal transition (EMT) and cell metastasis [35]. Interestingly, the lower expression of miR-148a was positively correlated with TGF-BETA signaling (NES = 1.59, *p* value < 0.05, FDR q < 0.2) (Figure 7B) and TGF-β treatment significantly inhibited miR-148a expression levels (Figure 7C). To determine whether miR-148a/PDK1 was involved in the EMT process, we found that the expression of the EMT-related protein Vimentin was greatly decreased and E-cadherin was increased in cells overexpressing miR-148a compared to the control, while PDK1 forced expression reversed the effect of miR-148a-inhibited EMT (Figure 7D). Similarly, the immunofluorescence (IF) results also showed that PDK1 forced expression reversed the effects of Vimentin and E-cadherin expression affected by miR-148a overexpression (Figure 7E). We carried out Transwell assays to evaluate the migration ability of miR-148a and PDK1 in the breast cancer cells, and showed that PDK1 counteracted the inhibitory effect of miR-148a on cell migration ability (Figure 7F). These results indicate that miR-148a targeting PDK1 inhibited EMT and cell migration in breast cancer cells.

### 3.8. PDK1 Partially Restores the Inhibitory Effect of miR-148a on Tumor Growth

To investigate the effects of miR-148a/PDK1 in vivo, the stable cell lines of MDA-MB-231/miR-NC, MDA-MB-231/miR-148a, and MDA-MB-231/miR-148a + PDK1 were subcutaneously injected into BALB/c mice. Tumor sizes were measured starting initially after 10 days, and the results showed that miR-148a overexpression significantly inhibited tumor growth, and this effect could be reversed by PDK1 overexpression (Figure 8B). After three weeks, xenograft tumors were harvested and weighed. We found significantly smaller tumor sizes and weight in the miR-148a overexpression group as compared with miR-NC, and the PDK1 overexpression in miR-148a partially restored tumor growth (Figure 8A,C). The tumor tissue immunohistochemistry (IHC) results showed lower PDK1 expression levels in the MDA-MB-231/miR-148a group than the miR-NC group (Figure 8D). The PDK1 overexpression group could reverse Vimentin and Bcl-2 expression and showed a much deeper staining than the miR-148a group (Figure 8D,E). These results suggest that miR-148a inhibited tumor development through PDK1.

## 4. Discussion

PDK1 is an important regulator for the Warburg effect, primarily regulating pyruvate dehydrogenase (PDH) activity which is induced in human cancers and represents a plausible anticancer therapeutic strategy [36]. In our study, we found that PDK1 expression levels were significantly higher in breast cancer tissues and cell lines. Histone modification is considered an important transcriptional regulatory mechanism and is involved in the progression of multiple tumors, for example, HDAC3-mediated deacetylation leads to ENO2 activation in pancreatic cancer [37], and the HDAC4-RelB-p52 complex regulates multiple myeloma growth [38]. Here, we found that HDAC2 and EZH2 were involved in the glycolysis pathway and had a positive correlation with PDK1 expression, which suggests that a regulatory relationship could exist between HDAC2/EZH2 and PDK1.

miRNAs are known to be associated with tumor cell proliferation, apoptosis, metastasis, and chemotherapy [39,40]. Notably, miR-148a is a member of the miR-148/152 family, and acts as a tumor suppressor in a variety of human solid tumors and participates in biological functions by targeting mature mRNAs, including WNT-1, Bcl-2, and KLF6 [41,42,43]. In our previous study, we demonstrated that miR-148a directly regulated PKM2 to affect the progress of aerobic glycolysis [44]. Accumulating evidence has indicated that aerobic glycolysis is important in tumor growth, tumor metastasis and chemoresistance [45,46]. In this study, we further found that miR-148a directly targeted PDK1 to inhibit its expression. It has been reported that PDK1 enhances EMT to contribute to the cisplatin resistance of ovarian cancer [33]. In our study, we demonstrated that miR-148a increased breast cancer cells’ sensitivity to Adriamycin through targeting PDK1. In addition, we found that TGF-β significantly inhibited miR-148a expression, while PDK1 could reverse the repression of the EMT process by miR-148a. It has been reported that TGF-β regulates EZH2 expression by upregulating Sox4 [47]. Therefore, we speculate that TGF-β may regulate miR-148a/PDK1 through EZH2, although this needs further study in the future.

Recently, several studies have indicated that the inhibition of HDACs and EZH2 is involved in miRNA dysregulation in tumors [48]. In nasopharyngeal carcinoma, miR-4465 expression can be modulated by histone deacetylase 7 (HDAC7) [49]. Myc acts as a repressor of miRNA-29 by recruiting histone deacetylase 3 (HDAC3) and EZH2 in aggressive B-Cell lymphomas [30]. In our study, we demonstrated that HDAC2 and EZH2 induced the dysregulation of miR-148a expression through epigenetic regulation. More specifically, HDAC2 and EZH2 can combine to form an inhibitory complex and repress miR-148a expression. Our previous study demonstrated that the overexpression of DNMT1 led to the hypermethylation of the miR-148a promoter region [29]. Interestingly, we innovatively found that DNMT1 recruited EZH2 and HDAC2 to the miR-148a promoter region to suppress the expression of miR-148a. In summary, these results indicate that histone modification participates in the regulation of PDK1 by inhibiting miR-148a, and the existence of the HDAC2/EZH2/miR-148a/PDK1 regulatory axis may be important in breast cancer development and therapeutic resistance (Figure 8F). Taken together, our study expounds a new mechanism axis to promote the development of breast cancer and treatment resistance. These key molecules can be used as potential biological targets for clinical diagnosis. Combined with EZH2 inhibitors and traditional chemotherapy, this will provide new ideas and directions to counter chemotherapy resistance in breast cancer.

## 5. Conclusions

In the present study, we demonstrated that PDK1 is a new direct target of miR-148a in mediating the development of breast cancer and Adriamycin resistance. HDAC2 and EZH2 upregulated PDK1 expression by silencing miR-148a expression. To identify the mechanism of regulation, we demonstrated that HDAC2 and EZH2 were recruited into the promoter region of miR-148a by DNMT1. This study found a new mechanism to regulate coordination between epigenetics (histone and miRNA) modification and glucose metabolism, which contribute to the progress and Adriamycin resistance of breast cancer, and demonstrated that the HDAC2/EZH2/miR-148a/PDK1 axis mediates breast cancer development and therapeutic resistance.

## Figures and Tables

**Figure 1 cancers-14-03600-f001:**
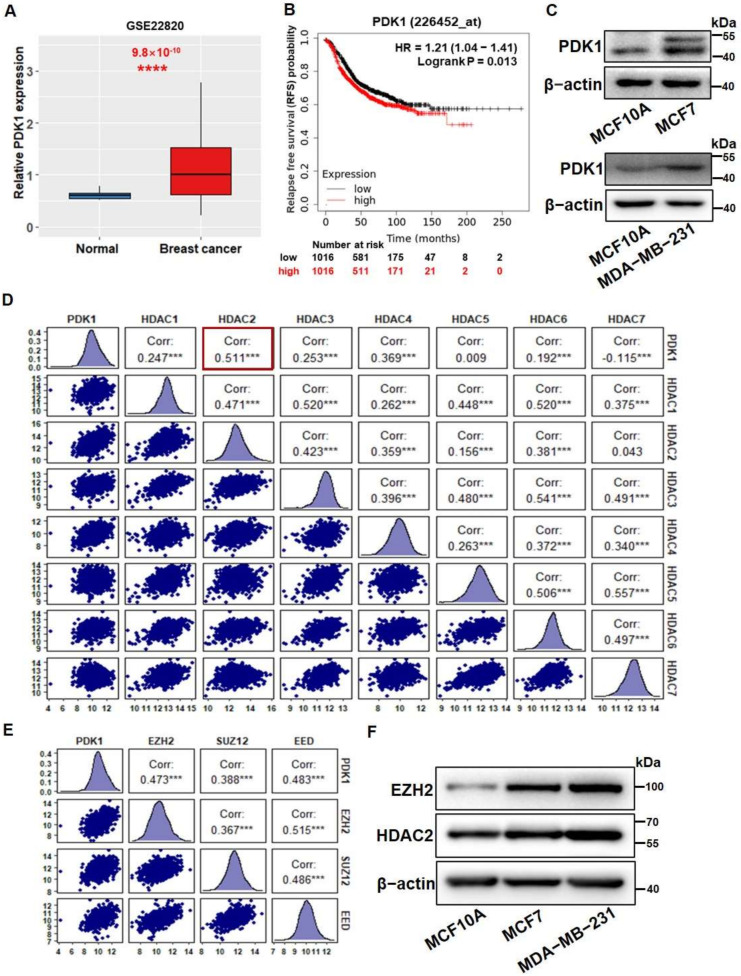
PDK1 levels were upregulated by HDAC2 and EZH2, which are associated with glycolysis. (**A**) The expression levels of PDK1 were analyzed in normal and breast cancer tissues from GEO dataset (GSE22820). (**B**) The relapse-free survival time was analyzed in groups with high and low expression of PDK1. (**C**) The protein levels of PDK1 were detected by Western blotting in MCF10A, MCF7, and MDA-MB-231 breast cancer cell lines. (**D**) The Pearson’s correlation analysis was conducted to test the correlation between levels of PDK1 and HDACs, including HDAC1-7. The data distribution, correlation coefficients between genes, and statistical significance were shown; the most significant correlation is indicated with a red border. (**E**) The Pearson’s correlation analysis was used to verify the correlation between PDK1 and PRC2, including EZH2, SUZ12, and EED levels. (**F**) HDAC2 and EZH2 protein levels were detected by Western blotting (**G**–**I**). The GSEA program was employed to analyze HALLMARK_GLYCOLYSIS pathway enrichment scores between (**G**) PDK1 high- and low-expression groups, (**H**) HDAC2 high- and low-expression groups, and (**I**) EZH2 high- and low-expression groups in TCGA BRCA dataset. (**J**) The protein expression levels of PDK1 were measured by Western blotting in the MDA-MB-231 control group (siNC), HDAC2 silencing group (siHDAC2), and EZH2 silencing group (siEZH2). Using the median value as the grouping standard. *** Indicates significant difference at *p* < 0.001, **** indicates significant difference at *p* < 0.0001. Original Blots see Appendix A.

**Figure 2 cancers-14-03600-f002:**
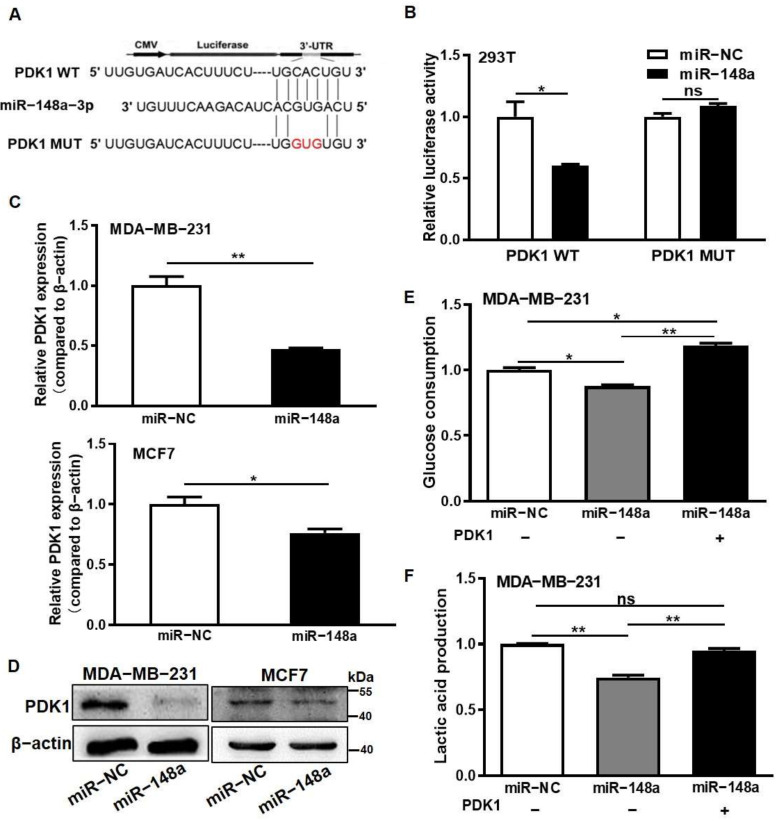
miR-148a was identified for directly suppressing PDK1 expression and inhibiting glycometabolism. (**A**) The PDK1 3′-UTR wild-type (WT) and mutant (MUT) reporter constructs containing miR-148a-targeting sequences are shown. (**B**) PDK1 WT and MUT 3′-UTR reporter plasmids were constructed, and Firefly/Renilla luciferase activities of the miR-NC and miR-148a groups were measured in the cells by dual-luciferase reporter assay. (**C**) The mRNA levels of PDK1 were detected in miR-148a overexpression cells by qRT-PCR with β-actin as a control. (**D**) The protein levels of PDK1 were measured by Western blotting. (**E**,**F**) The cultured media from the MDA-MB-231 miR-NC, miR-148a, and miR-148a + PDK1 groups were collected for the measurement of glucose consumption and lactate production levels, as described in the Method section. Data were analyzed by the Student’s *t*-test from three independent experiments in (**B**–**D**). * Indicates significant difference at *p* < 0.05, ** indicates significant difference at *p* < 0.01, and ns means no significant difference. Original Blots see Appendix A.

**Figure 3 cancers-14-03600-f003:**
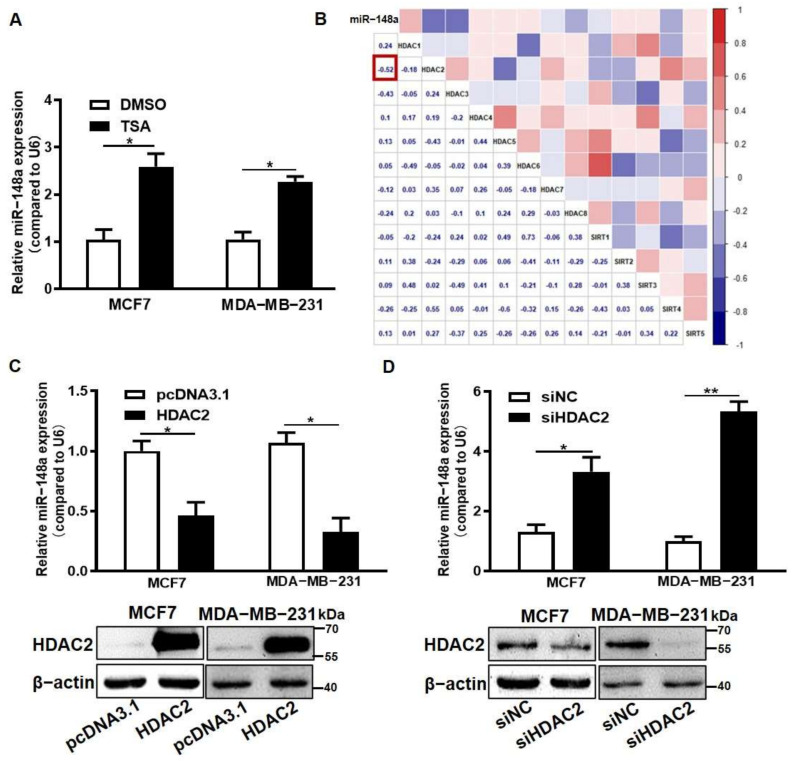
HDAC2 overexpression suppressed miR-148a expression. (**A**) MCF7 and MDA-MB-231 cells were treated with TSA or DMSO for 72 h. The expression levels of miR-148a were detected by qRT-PCR. (**B**) The correlations between miR-148a and histone acetylation modification-related proteins were analyzed (correlation: Pearson). (**C**,**D**) HDAC2 overexpression plasmid or HDAC2 siRNA were transfected into MCF7 and MDA-MB-231. After 48 h, miR-148a expression levels were detected in HDAC2 overexpression (**C**) or inhibition (**D**) groups by qRT-PCR. HDAC2 expression levels were detected by Western blotting. Data are shown as means ± SEM of three independent experiments in (**A**,**C**,**D**). Student’s *t*-test, * indicates significant difference at *p* < 0.05, and ** indicates significant difference at *p* < 0.01. Original Blots see Appendix A.

**Figure 4 cancers-14-03600-f004:**
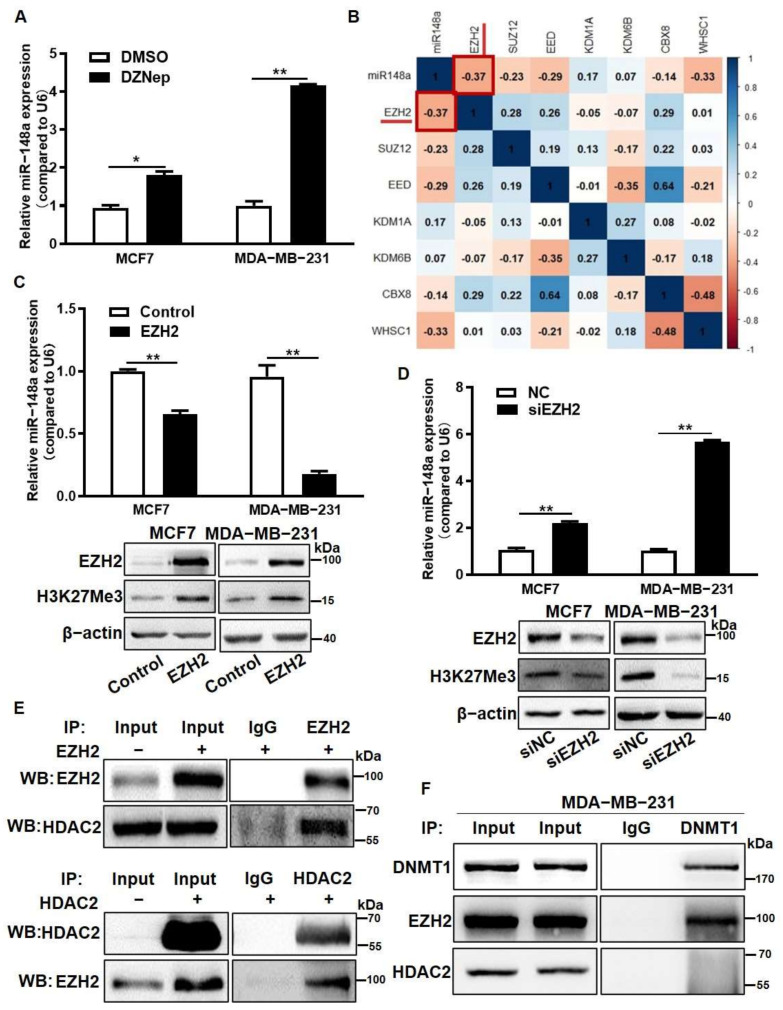
EZH2 overexpression inhibited miR-148a expression by recruiting HDAC2. (**A**) The miR-148a expression levels were detected in DZNep-treated MCF7 and MDA-MB-231 cells by qRT-PCR. (**B**) The correlation analysis between miR-148a and histone methylation modification-related proteins (correlation: Pearson). (**C**,**D**) The expression levels of miR-148a (upper) and the protein expression levels of H3K27Me3 (lower) were detected in EZH2 overexpression (**C**) or inhibition (**D**) groups by qRT-PCR and Western blotting (three independent experiments and each performed for three times). (**E**) Control, HDAC2, and EZH2 plasmids were overexpressed in MDA-MB-231 cells. Co-immunoprecipitation (Co-IP) assay was performed after 48 h, and Western blotting assay was used to measure EZH2 and HDAC2. (**F**) Co-IP assay was carried out in MDA-MB-231 cells. The levels of DNMT1, EZH2 and HDAC2 were detected in the input, IgG, and IP groups by Western blotting. Student’s *t*-test was used for comparison between two groups in (**A**,**C**,**D**). * Indicates significant difference at *p* < 0.05 and ** indicates significant difference at *p* < 0.01. Original Blots see Appendix A.

**Figure 5 cancers-14-03600-f005:**
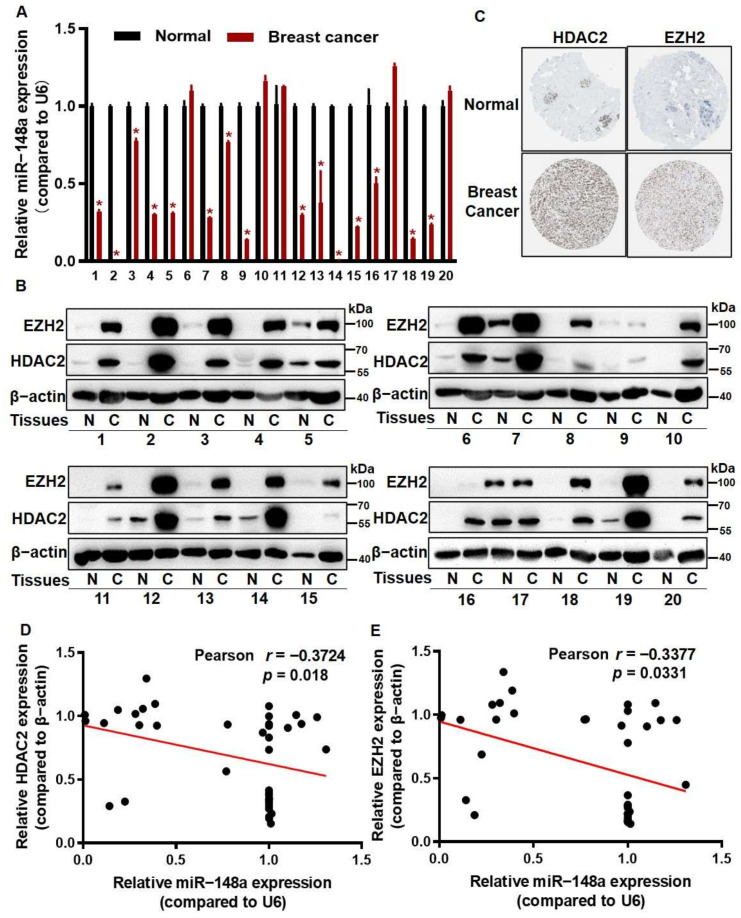
The levels of HDAC2 and EZH2 were inversely correlated with miR-148a levels in breast cancer tissues. (**A**) The expression levels of miR-148a were detected in 20 pairs of tissues by qRT-PCR. (**B**) EZH2 and HDAC2 protein expression levels were measured in 20 pairs of human breast cancer tissues and their corresponding normal breast tissues (N) by Western blotting. (**C**) The results from Human Protein Atlas showed a stronger staining of HDAC2 and EZH2 in breast cancer tissues than in normal tissues. (**D**,**E**) Pearson’s correlation analysis between miR-148a and HDAC2 (**D**) or EZH2 (**E**) in 20 pairs of tissues (HDAC2, *r* = −0.3724, *p* = 0.018) (EZH2, *r* = −0.3377, *p* = 0.0331). For human tissue samples, Student’s paired two-tailed *t*-test was used to analyze the comparison between different groups. * Indicates significant difference at *p* < 0.05. Original Blots see Appendix A.

**Figure 6 cancers-14-03600-f006:**
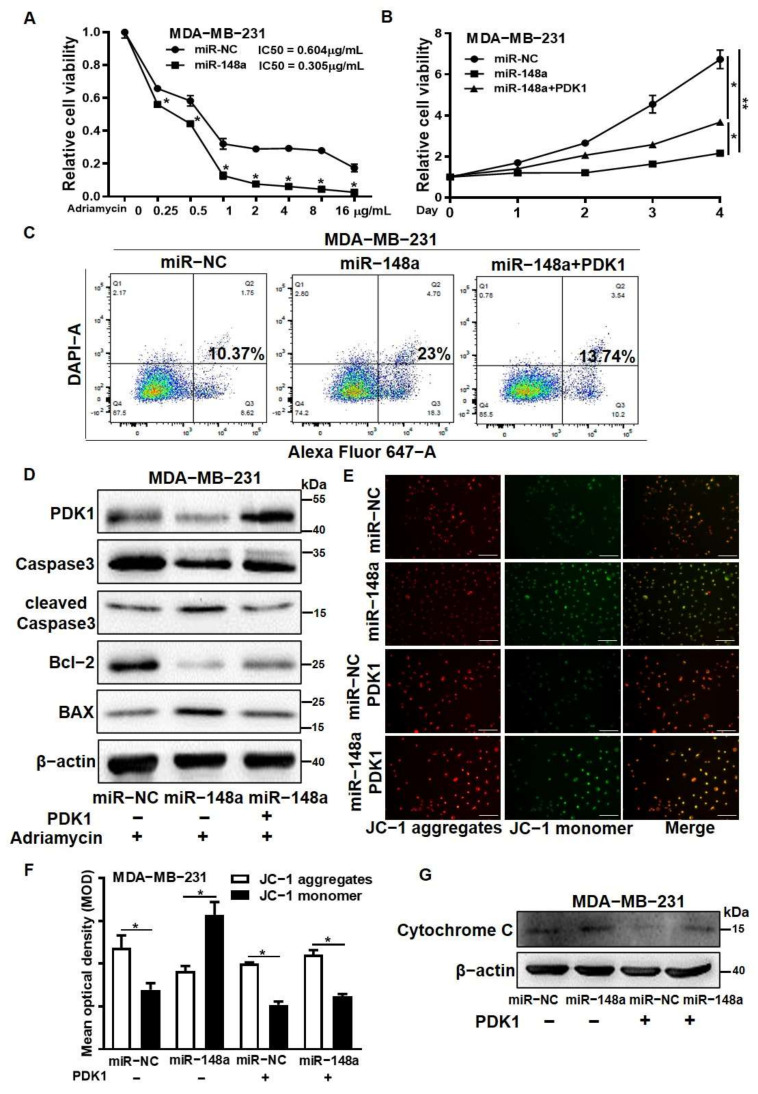
PDK1 overexpression reversed the effect of miR-148a decreasing Adriamycin resistance. Co-transfection of miR-148a and PDK1 lentivirus plasmid was performed in MDA-MB-231 cells to obtain double-overexpression stable cells. The following experiments were performed in the MDA-MB-231 miR-NC, miR-148a, and miR-148a + PDK1 groups. (**A**) The IC50 values of the MDA-MB-231 miR-NC and miR-148a groups treated with Adriamycin (0–16 μg/mL) were measured using CCK8 assays (*n* = 4). (**B**) The proliferative abilities of MDA-MB-231 miR-NC and miR-148a cells were detected under Adriamycin treatment (0.15 μg/mL) by CCK8 assays (*n* = 4). (**C**) Flow cytometry was used to measure the apoptosis levels in the three groups under Adriamycin treatment (0.3 μg/mL). (**D**) Western blotting was used to detect the protein expression levels of BAX, Bcl-2, cleaved Caspase3, Caspase3, and PDK1 in MDA-MB-231 cells treated with 0.15 μg/mL Adriamycin (three independent experiments performed in triplicate). (**E**) Mitochondrial membrane potential (MMP) changes were measured using the fluorescent probe JC-1 in the four groups. Representative immunofluorescence images (200×) are shown. Red represents JC-1 aggregates and green represents JC-1 monomer. (**F**) The results in (**E**) were quantified by calculating the image mean optical density (MOD). (**G**) The levels of Cytochrome C in cytoplasm were measured by Western blotting. Student’s *t*-test was used for comparison between two groups, bar = 50 μm. * Indicates significant difference at *p* < 0.05 and ** indicates significant difference at *p* < 0.01. Original Blots see Appendix A.

**Figure 7 cancers-14-03600-f007:**
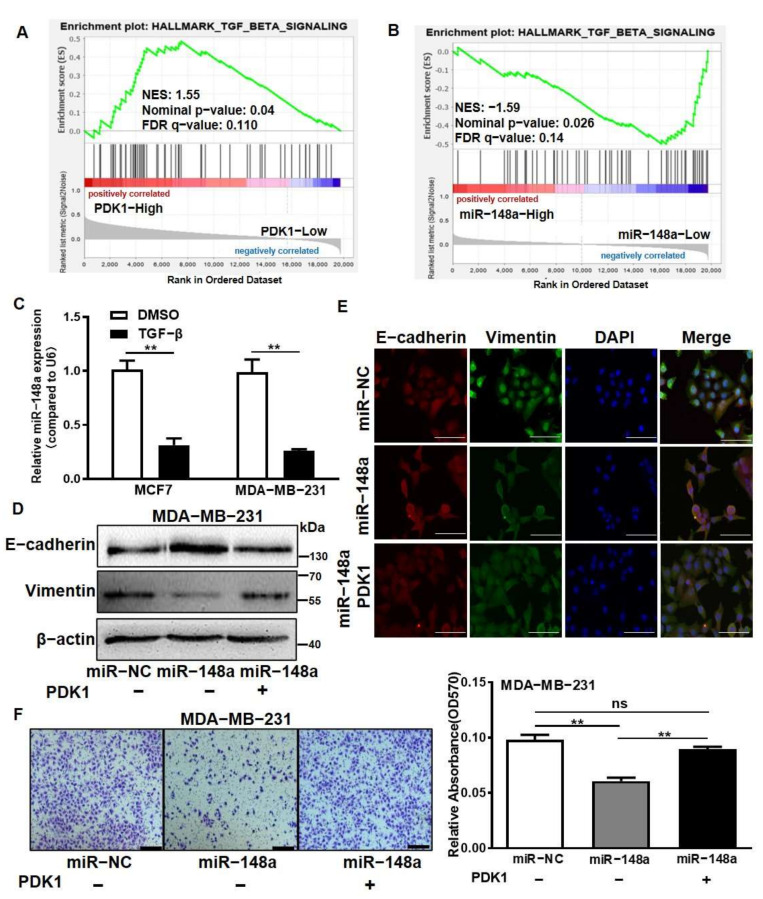
PDK1 suppression by miR-148a inhibited epithelial–mesenchymal transition and cell migration. (**A**,**B**) The GSEA program was employed to analyze KALLMARK_TGF_BETA_SIGNALING pathway enrichment scores between PDK1 (**A**) and miR-148a (**B**) high- and low-expression groups using TCGA BRCA dataset. (**C**) TGF-β was used to treat MCF7 and MDA-MB-231 cells, and the expression levels of miR-148a were detected by qRT-PCR. (**D**) The protein levels of E-cadherin and Vimentin were measured in MDA-MB-231 miR-NC, miR-148a, and miR-148a + PDK1 groups by Western blotting. (**E**) Immunofluorescence assay (IF) was used to detect E-cadherin (red) and Vimentin (green) protein levels in MDA-MB-231 miR-NC, miR-148a, and miR-148a + PDK1 groups. Representative images (400×) are shown (bar = 50 μm). (**F**) The invasiveness abilities of MDA-MB-231 miR-NC, miR-148a, and miR-148a + PDK1 cells were assessed by Transwell assays. Representative images (200×) are shown (left). The numbers of invaded cells were quantified by detecting the relative absorbance of crystal violet at OD570. Three independent experiments were performed in (**C**–**F**). Student’s *t*-test was used for comparison between two groups in (**C**,**F**). ** Indicates significant difference at *p* < 0.01 and ns denotes no significance. Original Blots see Appendix A.

**Figure 8 cancers-14-03600-f008:**
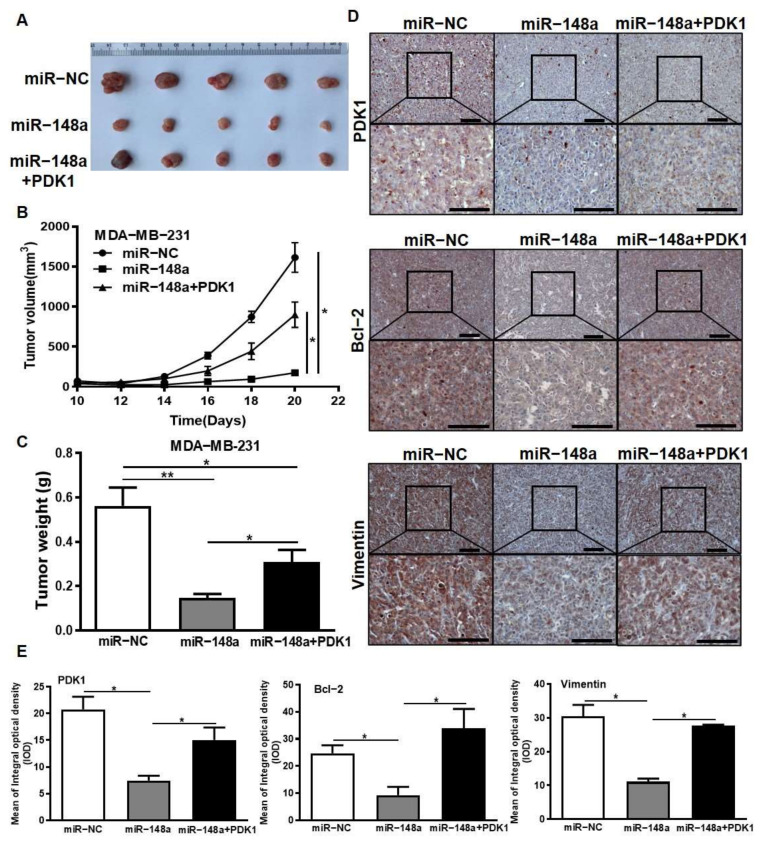
PDK1 partially restores the inhibitory effect of miR-148a on tumor growth. (**A**) MDA-MB-231 carrying miR-NC, miR-148a, and miR-148a + PDK1 stable cells (5 × 10^6^ cells) were subcutaneously implanted into female BALB/c mice. These tumors were harvested after three weeks. (**B**) Tumor growth curves were plotted. Tumor volume calculation formula: volume = 0.5 × Length × Width^2^. (**C**) The weight of harvested tumors was analyzed. (**D**) The protein expression levels of PDK1, Bcl-2, and Vimentin were examined in harvested tumors tissues by IHC assay (Upper: 200×; Lower: 400×; bar = 50 μm). (**E**) The IHC results were quantified by Image-Pro Plus 6.0 software. (**F**) The pattern diagram. Tumor volumes were compared using analysis of variance of repeated measurement. Student’s *t*-test was used for comparison between two groups in (**C**,**E**). * Indicates significant difference at *p* < 0.05 and ** indicates significant difference at *p* < 0.01.

## Data Availability

The datasets generated/analyzed during the current study are available.

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
