# Peer review of "HDAC2- and EZH2-Mediated Histone Modifications Induce PDK1 Expression through miR-148a Downregulation in Breast Cancer Progression and Adriamycin Resistance"

_cancers, 2022, doi:10.3390/cancers14153600_

Round 1

Reviewer 1 Report

The authors demonstrated the role of HDAC2/EZH2/miR-148a/PDK1 axis in breast cancer. Its approach and results are novel and interesting, however, some point should be revised. The specific comments are as following:
Abstract: Introduction and Methods parts were lengthy.

Introduction: line 62, error (ariginine~~). The goal of this study should be emphasized.

 Discussion: Compared to massive and interesting result, Discussion part was weak. Clinical value of these data should be presented more.

Reference: please check reference style.

Author Response

Comment 1: Abstract: Introduction and Methods parts were lengthy.

Answer: Thank you for your careful review of our manuscript. Introduction and Methods parts have been simplified. The revised content has been marked in the manuscript.

Comment 2: Introduction: line 62, error (ariginine~~). The goal of this study should be emphasized.

Answer: We have corrected the error in revised manuscript and re-wrote the goal of this study to highlight the study's objective.

Comment 3: Discussion: Compared to massive and interesting result, Discussion part was weak. Clinical value of these data should be presented more.

Answer: We agree with the comment. To further convey the significance of our study, we have added more clinical value of these data in Discussion part.

Comment 4: Reference: please check reference style.

Answer: We have re-verified that the reference format complies with the magazine's standards.

Reviewer 2 Report

The authors found that epigenetic changes mediated by HDAC2 and EZH2 upregulate PDK1 through miR-148a suppression in breast cancer. DNMT1 bound directly to EZH3 and recruited EXZH2 and HDAC2 complex into the promoter region of miR-148a, leading downregulation of miR-148a.

 Recent studies have shown an association between miR-148a and poor survival of breast cancer, so the results of this study that revealed the HDAC2/EZH2/miR-148a/PDK1 axis will contribute to understanding the regulation of breast cancer development.

In Figure 7D, the authors showed that the expression of the EMT-related protein vimentin was significantly reduced and E-cadherin was increased in cells that overexpressed miR-148a compared to the controls. Furthermore, forced expression of PDK1 was able to reverse the EMT inhibitory effect of miR-148a. Does this mean that forcedly expressed PDK1 promoted vimentin transcription but reduced E-cardherin transcription? The authors need to explain this possibility.

 Vimentin inhibition may be mediated by a decrease in PDK1 in miR-148a transfected cells. Overexpression of PDK1 in cells transfected with this miR-148a may promote vimentin transcription and increase cell invasion. On the other hand, MiR-148a has been shown to inhibit the migration and invasion of breast cancer by directly targeting WNT-1 (line 575, ref 41). Even in this case, PDK1 may also promote WNT-1 transcriptionally, causing upregulation of vimentin. Therefore, the PDK-1 knockdown experiment can help determine these possibilities. In the first case, overexpression of miR-148a does not affect vimentin in cells into which PDK1 siRNA has been introduced. However, in the latter case, miR-148a expression will increase vimentin expression in the absence of PDK1. It is necessary to discuss the possibility that miR-148a acts directly on WNT-1 and PDK1.

 Post-transcriptional inhibition of bcl-2, an anti-apoptotic factor, by miR-148a has also been reported (line 575, ref 42). Transfection of miR-148a may directly reduce bcl-2 expression in the absence of PDK1. This also need to be discussed.

 Figure 6B: Tumor cell growth in miR-148a group was slower than that of the miR-NC group, even when intracellular PDK1 levels were restored to control levels by overexpression of PDK1 (lane 1 and lane 3 in Figure 6D). This incomplete recovery of tumor cell growth was also observed in vivo studies (Figure 8B). It is necessary to explain why overexpression of PDK1 did not restore tumor cell growth and tumor growth.

Reviewer 3 Report

this study looked into the effect of HDAC and EZH2 expression on PDK1 expression in breast cancer. Although I am convinced by the results of HDAC/EZH2 regulated PDK1 expression, so of the results need to be fixed. This includes Figure 1 J- western blot showing HDAC knockdown not very strong - would like to see a better western blot. In addition, Figure 7 for the western- can you include other EMT factors such as Twist, snail , slug - at least one or two would be convincing. 

Reviewer 4 Report

In this manuscript, Xie et al elucidated how epigenetic regulation of PDK1 (pyruvate dehydrogenase kinase 1) mediates multiple mechanisms that promote tumor progression. The study is comprehensive testing both in vitro and in vivo models. Some technical issues should be addressed. The kind of antibodies used in immunoassays was not indicated (polyclonal or monoclonal, as well as clone number, which is more informative than catalog numbers). The authors need to include the flow cytometry procedure because they evaluated apoptosis but the methodology was not included. Western blot bands need to include molecular weight markers. Although the authors mention that tumor samples were obtained from a tissue bank, they need to include some demographics about the sample population (age, type of tumor (luminal, TNC, etc), stage). Changes in monomer formation, reported in Fig 6, could be a consequence of apoptosis, as reported in Fig 6C; thus, the authors need to include a viability marker to discard this possibility. 

Round 2

Reviewer 5 Report

This manuscript is a resubmission of an earlier submission. The following is a list of the peer review reports and author responses from that submission.